# Sodium Salicylate as Feed Additive in Broilers: Absence of Toxicopathological Findings

**DOI:** 10.3390/ani13091430

**Published:** 2023-04-22

**Authors:** Mayra Carraro Di Gregorio, Elaine Renata Motta de Almeida, Claudia Momo, Cristiane Soares da Silva Araújo, Isis Machado Hueza, Newton Andréo-Filho, Leonila Ester Reinert Raspantini, André Tadeu Gotardo, Silvana Lima Górniak

**Affiliations:** 1Research Centre for Veterinary Toxicology (CEPTOX), Department of Pathology, School of Veterinary Medicine and Animal Science, University of São Paulo, Pirassununga 13635-900, Brazil; mayracdg@usp.br (M.C.D.G.); erm.almeida@usp.br (E.R.M.d.A.); cmomo@usp.br (C.M.); imhueza@unifesp.br (I.M.H.); esterraspantini@usp.br (L.E.R.R.); andregotardo@usp.br (A.T.G.); 2Department of Nutrition and Animal Production, School of Veterinary Medicine and Animal Science, University of São Paulo, Pirassununga 13635-900, Brazil; crsilva@usp.br; 3Department of Pharmaceutical Sciences, Institute of Environmental, Chemical and Pharmaceutical Sciences, Federal University of São Paulo, Diadema 09913-030, Brazil; newton.andreo@unifesp.br

**Keywords:** non-steroidal anti-inflammatory drugs, prolonged administration, poultry, growth promoter alternative

## Abstract

**Simple Summary:**

The use of antimicrobial growth promoters (AGPs) in animal production has been associated with the increase in bacteria resistance to multiple drugs. The ban on the use of AGPs in many countries has highlighted the increasing need for alternatives as non-antibiotic growth promoter feed additives. Thus, a study was conducted on broiler chickens to test the effectiveness of sodium salicylate (a non-steroidal anti-inflammatory drug) as a non-antibiotic anti-inflammatory agent. The results showed that sodium salicylate did not have a significant effect on the health of animals; however, the results suggest that further studies may be necessary under more stressful conditions to better evaluate the efficacy of these agents as growth promoters.

**Abstract:**

Antimicrobial growth promoters (AGPs) in animal production have been related to the increase in multidrug-resistant bacteria. The AGP ban in many countries has highlighted the growing need for alternatives for feed additives. Considering the non-antibiotic anti-inflammatory theory of AGPs, chicks received three different doses of sodium salicylate (SS) in feed (10, 30, 90 mg/kg), basal diet (BD) was used as a negative control, and zinc bacitracin (ZB) was used as a positive control. Chicks were individually housed to increase the accuracy of the dose of SS ingested. Performance parameters and footpad dermatitis were evaluated weekly, while haematology, serum biochemistry, histopathology, and tibial dyschondroplasia were determined on Days 21 and 42. A linear dose-dependent decrease in haemoglobin concentration was observed, but the values were within the normal reference range. Among all the other evaluated parameters, no relevant differences between treatments were observed; however, not even the AGP group performed better than the control group. It is possible that the conditions in which the birds were raised were not stressful enough to allow for anti-inflammatories to demonstrate their beneficial effects on performance. Studies should be conducted where the animals are exposed to commercial conditions, as the presence of natural stressors could allow a better evaluation of the efficacy of the anti-inflammatory agent as a growth promoter.

## 1. Introduction

Poultry is the main animal protein source for human nutrition, with a global production of more than 100 million tons in 2021 [1]. For decades, the growth of poultry production was supported by the administration of non-therapeutic concentrations of antimicrobial agents called antimicrobial growth promoters (AGPs) [2,3]. This feed additive was given at a low dose to improve the growth rate and feed conversion efficiency, and to reduce the cost of production [3]. However, there is worldwide concern related to the potential of AGPs to generate and select multidrug-resistant bacterial strains, as well as the presence of antibiotic residues in meat products and environmental contamination [4]. In fact, the World Health Organization (WHO) has recommended the prohibition of antimicrobial agents that are administered as AGPs [5]. Recently, the Food and Agriculture Organization of the United Nations (FAO) described antimicrobial resistance as the quintessence of the One Health issue [6]. As a result, AGPs have been banned in many countries [7,8].

Banning AGPs without replacement options will result in reduced animal performance and an increased incidence of animal disease [4], which also increases the use of therapeutic antibiotics and severely impacts the productive chain of chicken meat [9]. In this way, the survival of poultry production is dependent on the balance between feed costs, disease control, and meat yield [4]. Some compounds have been described as alternatives, such as probiotics, prebiotics, symbiotics, organic acids, enzymes, and phytochemicals [8]. However, to date, no compound has been able to match the results that can be achieved with AGPs.

To develop effective non-antibiotic alternatives, it is first necessary to understand the biological mechanism of action of AGPs. There are two main theories, one of which posits that AGPs have antibacterial action on the intestinal microbiota, reducing subclinical infections and improving animal performance [3,10]. The second suggests that AGP action is related to the anti-inflammatory effect, working much more as growth permitters than growth promoters. In this case, AGP acts by direct inhibition of the intestinal inflammatory response and thinning of the intestinal wall, which facilitates the absorption of nutrients, decreases the production and excretion of catabolic mediators by intestinal inflammatory cells, which modify the microbiota and decreases muscle catabolism and anorexia [11,12].

In birds, non-steroidal anti-inflammatory drugs, especially sodium salicylate (SS) and acetylsalicylic acid (ASA), are frequently used due to their anti-inflammatory and analgesic properties [13]. These compounds have demonstrated their effectiveness in heat stress, locomotor disorders, and respiratory and digestive problems, as they improve welfare [14,15].

Thus, we hypothesized that the prolonged use of SS would improve the growth performance of broiler chickens by decreasing the inflammatory process in the gastrointestinal tract. To test this hypothesis, in the present study, we aimed to evaluate the safety of SS individually applied during all cycles of the production of broiler chickens.

## 2. Materials and Methods

### 2.1. Pharmacotechnics

The sodium salicylate (SS) used was grade PA (Dinâmica, Indaiatuba, Brazil). An assay was carried out to determine the saturation solubility coefficient (shake-flask method) of SS in ethanol, methanol, and acetonitrile. For this, SS, in sufficient quantity to saturate the system, was added to 20 mL of solvent and stirred for 24 h. Afterwards, an aliquot of the supernatant was filtered in PTFE, hydrophilic 0.22 µm, and diluted 1:100, and the reading was performed using UV spectrophotometry using the 297 nm wavelength filter (Evolution 201 UV visible spectrophotometer, Thermo Scientific, Waltham, MA, USA). Assays were performed in triplicate.

The incorporation of the drug was carried out via cornmeal, adding the drug and water until the final concentration was 5.12% SS. The high viscosity conferred on the wet mass by the SS prevented its inclusion at higher concentrations. Afterwards, the granules were manually prepared using a 1.18 mm sieve and kept in a forced circulation oven (model 420-1DE, Nova Ética - Ethik Technology, Vargem Grande Paulista, SP, Brazil) at 45 °C for 7 h. Residual moisture was monitored on an electronic moisture scale (Model MAC210, Radwag Ltd., Radom, Poland), where 0.5 g of the granulate was evaluated in triplicate, and the process ended when the residual moisture was equal to or less than 5%.

The calibration curve was obtained through serial dilution in methanol of the SS standard containing 62.70 µg/mL in 6 levels (62.70; 31.35; 15.67; 7.84; 3.92; 1.96) in triplicate. The reading was performed by UV spectrophotometry (Evolution 201 UV visible spectrophotometer, Thermo Scientific, Waltham, MA, USA), obtaining R^2^ = 0.9948.

Briefly, the extraction procedure was performed by adding 50 mL of methanol to 20 mg of granules, vortexed for 30 s, sonicated for 15 min, and left to rest for 2 h. Subsequently, the solution was again vortexed, sonicated, and centrifuged at 5000 RPM for 10 min (20 °C). The supernatant was analysed by UV spectrophotometry. The recovery of SS was 106.7%, and the coefficient of variation was 1.1%.

### 2.2. Birds, Feeding and Housing

A total of 70 one-day-old Cobb 500 male broiler chicks vaccinated against Marek and Gumboro diseases were purchased from a commercial hatchery and were used. The experiment was performed in the experimental houses of the Poultry Research Laboratory of the Department of Animal Nutrition and Production (School of Veterinary Medicine and Animal Science, University of São Paulo, Pirassununga, Brazil).

Until the 21st day of life, chicks were individually housed in stainless steel vertical batteries (100 × 34 × 24 cm; l, w, h) equipped with trough-type feeders, nipple drinkers, and concealers. On the 22nd day, the birds were transferred to galvanized steel wire cages (50 × 50 × 45 cm; l, w, h) equipped with nipple drinkers and with controlled temperature and photoperiod. Animals were maintained at a comfortable temperature according to their age, with feed and water ad libitum, and a controlled photoperiod.

Diets were based on corn and soybean meal formulated for a two-phase feeding program (starter: d 0–21, and grower: d 22–42) to meet the nutritional and energy recommendations suggested by Rostagno et al. [16] (Table 1).

### 2.3. Experimental Design

Broilers were randomly allocated into five treatment groups (*n* = 14 per group): (1) negative control, which received only the basal diet (BD group); (2) positive control, which received the basal diet supplemented with 55 ppm zinc bacitracin (ZB group); and three groups, which received the basal diet supplemented with SS granules (5.12%) at doses of 10, 30, or 90 mg of sodium salicylate per day (mg/kg BW; S10, S30 and S90 groups, respectively). The doses used in this experiment were based on findings from a previous experiment [17].

To reach the intended doses, the inclusion of the drug in the feed was adjusted weekly. To determine the appropriate amount to be incorporated into feed, calculations were made for feed consumption/day in relation to the development curve of the broilers [18]. The cage with one broiler was considered the experimental unit, and each bird was monitored from the 1st to the 42nd day of life.

### 2.4. Performance and Blood Parameters

Feed intake (FI) and body weight (BW) were assessed weekly, and the performance variables body weight gain (BWG) and feed conversion ratio (FCR) were calculated for each period. The broilers were measured individually.

Blood samples were collected at the 21st and 42nd day from 12 and 9 birds per treatment, respectively, by ulnar venipuncture. Blood samples were placed in K3 EDTA and serum separator clot activator plastic tubes (Vacuette, Greiner, Bio-One GmbH, Kremsmünster, Austria) for haematological and biochemical analysis.

The haematological evaluation was performed by a manual method in a Neubauer chamber using blood diluted on 0.01% toluidine blue stain at a 1:100 dilution. Differential white blood cell counts were made using an average of 100 cell counts from blood smears with Wright’s stain and examined under an optical microscope with a 100X objective. Haematocrit was determined in whole blood using a micro-haematocrit tube centrifuged at 11,500× *g* for 5 min, and the results were estimated as a percentage. Determination of haemoglobin concentration was performed by the cyanomethemoglobin method using the reagent from Bioclin^®^ (Quibasa Química Básica Ltda., Belo Horizonte, Brazil) and evaluated using a spectrophotometer (BTS 310, Biosystems, Recife, Brazil). Mean corpuscular haemoglobin (MCH), mean corpuscular haemoglobin concentration (MCHC), and mean corpuscular volume (MCV) were calculated.

Serum biochemistry was analysed using an automated biochemical analyser (ChemWell T, Awareness Technology Inc, Palm City, FL, USA) with Bioclin^®^ (Quibasa, Belo Horizonte, Brazil) according to the manufacturer’s instructions. Aspartate aminotransferase (AST), alanine aminotransferase (ALT), alkaline phosphatase (ALP), lactate dehydrogenase (LDH), gamma glutamyl transferase (GGT), total protein (TP), albumin (ALB), creatine kinase (CK), glucose (GLU), creatinine (CR), and uric acid (UA) were evaluated.

### 2.5. Histopathology

On the 21st and 42nd days, 5 and 6 broilers from each group, respectively, were euthanized, and samples from the crop, proventriculus, gizzard, small intestine (duodenum, jejunum, and ileum), caecum, liver, and kidneys were collected for histopathological evaluation. The samples were fixed in 10% buffered formaldehyde and processed according to routine protocols. Sections (5 µm) were stained with haematoxylin and eosin and examined under an optical microscope. Lesions were classified as mild, moderate, or severe according to their intensity and as focal, multifocal, or diffuse based on their distribution.

### 2.6. Tibial Dyschondroplasia and Foot-Pad Dermatitis Evaluation

The legs of the chickens were separated during slaughter and frozen at −20 °C until processing and analysis of tibial dyschondroplasia (TD). The macroscopic evaluation was performed on the right tibia samples, which were stripped and cut longitudinally. Measurements were performed using the ImageJ program (Fiji distribution, https://imagej.net/Fiji/Downloads, accessed on 1 March 2023). The growth cartilage thickening of these bones was evaluated, and scores were assigned according to the degree of injury: 0, no evidence of TD; 1, thickening between 1 and 3 mm; 2, thickening between 3.1 and 6.0 mm; and 3, thickening greater than 6 mm [19].

The observation of the integrity of the plantar pad was performed weekly on both pads of the birds and classified according to Martrenchar et al. [20]: 0, no injury; 1, injury to less than 25% of pads; 2, lesion between 25 and 50% of the pads; and 3, injury to more than 50% of pads.

### 2.7. Statistical Analysis

Individual chicks were considered the experimental unit for the measurements of BW, BWG, FI, feed conversion ratio, and haematological and biochemical parameters. All data were checked for outliers and normality of the residuals. Outliers and negative feed conversion values due to weight loss were removed from the analysed dataset and the Excel file in the repository (data accessibility). Statistical analysis was performed with SAS (version OnDemand for Academics) and analysed using the Generalized Linear Mixed Models procedure with the alpha value set at 0.05. Linear and quadratic polynomial contrasts were specified for dose–response modelling with correction for unequal spacing of SS doses, namely, 0, 10, 30, and 90 mg/kg per day. Dunnett’s multiple comparison test was used to compare the reference diet (zinc bacitracin group) against the basal diet and diets formulated using SS. Data subjects with repeated measurements from the same individuals over a period of time were subjected to the linear mixed model, with treatment, time, and the treatment x time interaction as fixed. In the case of multinomial (ordered) response distributions (histopathology, tibial dyschondroplasia, and pododermatitis), the cumulative logit function was used to compare all treatments. The possible existing correlations between the studied variables were verified through CORR procedure of SAS. 

## 3. Results

### 3.1. Body Weight, Body Weight Gain, Feed Intake and Feed Conversion Ratio

The effects of the five dietary treatments on BW, FI, and F:G are summarized in Table 2. Linear and quadratic responses, as well as the comparison of the reference treatment (ZB) with the groups treated with different doses of SS, were not significantly different at the end of the trial period; however, ZB significantly enhanced the BWG of broilers at week 4 compared to the S10 and S90 groups (*p* = 0.0397). ZB also promoted an increase in BW at week 5 compared to the BD (0) and S10 groups (*p* = 0.0131). In the first week, the feed intake and feed:gain ratio presented a quadratic response (*p* = 0.0114 and *p* = 0.0031, respectively). ZB also increased feed intake and the feed:gain ratio compared to the BD group (*p* = 0.0084 and *p* = 0.0002, respectively). At week 4, ZB again reduced feed intake compared to the BD group (*p* = 0.0005). The addition of SS promoted a linear reduction in consumption in the sixth week (*p* = 0.0147), but it was not sufficient to impact the periods from 22 to 42 days and the total period of the experiment. There was a significant interaction between time and treatment (*p* ≤ 0.05) observed for FI.

### 3.2. Haematological and Biochemical Parameters

The haemogram (Table 3) revealed a significant linear dose-dependent decrease in haemoglobin concentration at 21 (*p* = 0.0158) and 42 days (*p* = 0.0006) and a significant difference between ZB and S90 (Day 21 *p* = 0.0096; Day 42 *p* = 0.0074). This resulted in a significant linear reduction in MCH and MCHC on Day 42 (*p* = 0.0076 and *p* ≤ 0.0001, respectively) and a lower value for MCH and MCHC in the S90 group compared to ZB on Day 21 (*p* = 0.0327 and *p* = 0.0253) and MCHC on Day 42 (*p* ≤ 0.0001). MCH also presented a significant interaction between age and treatment (*p* = 0.0462). Regarding the leukogram, only the HL ratio showed a significant linear reduction on Day 21 (*p* = 0.0272). All other parameters did not show significant linear or quadratic differences in relation to the reference group (ZB) or even the age–treatment interaction.

In serum biochemistry (Table 4) on Day 21, only LDH activity was linearly reduced (*p* = 0.0499). At this age, the use of ZB was also able to significantly reduce LDH compared to the BD group (*p* = 0.0384). At 42 days, significant linear decreases in ALT (*p* = 0.0472) and AST (*p* = 0.0111) and significant linear increases in TP (*p* = 0.0019), GLOB (*p* = 0.0022) and GLU (*p* = 0.0142) were observed. On the 42nd day, ZB significantly affected TP compared to the BD group (*p* = 0.0024) and GLU compared to the S90 group (*p* = 0.0220). Only ALP presented a significant interaction between age and treatment (*p* = 0.0106).

### 3.3. Histopathological Evaluation

In the histopathological evaluation of the small and large intestines, liver, and kidney, the presence of mild to moderate inflammatory infiltrate was verified in the groups treated with SS and in the BD and ZB groups; however, none of the changes showed a significant difference between treatments (data not shown). Although the histopathological evaluation showed no significant difference between the treatments, at 21 days, there was an increase in the average scores of renal mononuclear infiltrates according to the increase in the SS dose. This finding had a significant correlation with the BW (r = 0.46, *p* = 0.0350), WG (r = 0.44, *p* = 0.0499), and F:G ratio (r = −0.48, *p* = 0.0319) of the birds; however, these correlation patterns were not significant at 42 days.

Furthermore, none of the characteristic adverse effects of nonsteroidal anti-inflammatory drugs (NSAIDs), such as erosive damage or upper gastrointestinal bleeding, were observed in birds from any of the treatment groups.

### 3.4. Tibial Dyschondroplasia and Foot-Pad Dermatitis Evaluation

None of the animals across all treatment groups exhibited any lesions associated with pododermatitis or tibial dyschondroplasia (data not shown).

## 4. Discussion

The gastrointestinal tract has the difficult job of striking a balance between being an effective barrier and being selectively permeable to nutrients and tolerant of the associated resident microbiota [21]. Intestinal health is fundamental for the efficient absorption of nutrients and, consequently, for the well-being and health of animals. In poultry, intensive selection to improve daily weight gain and feed conversion ratio has generated breeds characterized by high feed intake. Beyond a certain threshold, excessive amounts of feed, as well as some ingredients, can place considerable stress on the digestive system and, even in the absence of a pathogen, can harm the health of the GI tract [22]. High antigenic stimulation leads to considerable mononuclear cell infiltration in the gut, giving rise to a state of constant controlled inflammation, also called “physiological inflammation” [23]. Thus, it is essential for health to control intestinal inflammation; for that purpose, important mechanisms are in place [24]. Inflammation linked to illness or feed is inversely related to growth and health, with the impact of inflammation on growth reduction dependent on the magnitude of the stimulus [25].

One of the main and most accepted theories about the growth-promoting effect of antimicrobial drugs is around their non-antibiotic anti-inflammatory effects [11,12]. Reinforcing this postulation, many studies have been conducted principally in birds and pigs [8,25,26,27,28]. In fact, proinflammatory immune responses have been associated with poor growth performance. This observation, when combined with studies that show the anti-inflammatory effect of antimicrobial agents administered in subtherapeutic doses, has led to suggestions that reducing the nutrient cost of gut inflammation may explain the growth-promoting or growth-permitting effect of AGPs [29]. In this context, we are interested in verifying whether SS has potential benefits for poultry welfare and performance. However, initially, it was necessary to verify the dosage of the anti-inflammatory that is actually being ingested by the bird conveyed through the feed, as well as to assess the possible adverse effects of SS at the doses employed here.

A search in the literature to provide data about the beneficial use of ASA supplementation in the diets of birds under different stress conditions [30,31,32,33,34,35] or not [36,37,38] revealed inconsistent results between the different studies, and no clear conclusion could be reached [15]. It should be noted that, in all these studies, authors have incorporated ASA into drinking water or feed; however, no descriptions about the condition and storage time, as well as the replacement of the feed/drinking water after incorporation of the salicylate until their consumption, were provided. On the other hand, the stability of the substance must be taken into consideration since, depending on the substrate and preparation in which the ASA is mixed, it can lead to its rapid degradation [39] and, consequently, to the bioavailability of the administered dosage. Indeed, we verified that the concentration of ASA in feed decreased significantly within 7 days during open storage in the presence of broiler feed [17]. Additionally, it was demonstrated that salicylic acid, a product of ASA hydrolysis, could potentiate adverse effects on broilers [40]. Thus, we chose to incorporate SS into the feed since free acid is more stable than ASA [40]. Situations such as heat stress and diseases can increase water consumption and/or decrease feed intake, so that the administered dose of drugs under these altered conditions may have been much higher or lower than the intended target dose. Indeed, for a more accurate assessment of adverse and toxic effects, it is better to know the dose (mg/kg of BW) rather than the concentration (mg/kg of feed) to avoid confusion with the actual dose [41]. Additionally, the birds were reared in individual cages, allowing consumption to be measured individually, providing the actual feed and SS intake per bird per period.

While the pharmacokinetics and pharmacodynamics of ASA and SS and their capacity to treat inflammation in birds are well known [13,42,43], data concerning the tolerance and/or toxic effects of prolonged salicylate use are lacking. Thus, we performed haematological and biochemical appraisals since they are primary tools used for evaluating the toxicity of analgesic and anti-inflammatory drugs [44]. In relation to the haematological evaluation, it is reported in the literature that the chronic use of SS in humans promoted a decrease in haemoglobin concentrations [45]. Goldstein et al. [46] also demonstrated that prolonged exposure to NSAIDs is associated with a significant decrease in haemoglobin and highlighted the issue of long-term chronic occult blood loss. Furthermore, Mohan et al. [47] demonstrated that relatively low doses of ASA (10 mg/kg/day) for five consecutive days in chickens caused a decrease in the number of erythrocytes, haemoglobin, and haematocrit, suggesting a haemorrhage from the GI tract and possibly anaemia. In the present study, although a significant linear reduction in haemoglobin was observed in animals treated with SS, at the 42nd day, the values were within the normal reference range [48]. No significant alterations in WBCs or differential leukocytes were detected between the different groups. These results corroborate the findings of Pòzniak et al. [14], who, even using higher doses of SS (200 and 400 mg/kg), did not observe changes in the leukogram due to the influence of SS.

In relation to the biochemical assessment, in the same manner as a previous study conducted by our group employing lower doses of SS (2.5, 5.0, and 10 mg/kg) in broilers for 42 days, we did not detect consistent differences in serum parameters evaluated [17].

Concerning histopathological evaluations, although studies have shown that ASA promotes lesions in the small intestine [38,47], lungs, liver, and kidneys [38], in the present study, no significant alterations were detected in any of the tissues evaluated. This result agrees with those of previous studies [17,49].

Environmental stresses, such as inadequate nutrition, crowding, moving and mixing of animals, poor sanitation, and high or low temperatures, contribute to increased responses to AGPs [11,50,51]. The reason behind the greater observation of this effect of AGPs is that under poor conditions the development of intestinal inflammation is favoured, and, therefore, the anti-inflammatory effects of the antimicrobial drug can be observed.

In the present study, we did not detect significant alterations in growth performance between the different groups at the end of the treatment. Thus, it is possible to suppose that in the same manner as AGP, anti-inflammatory drugs could show the greatest effects under stressful conditions. In fact, no changes were observed even in the animals of the positive control group, ZB, where an increase in weight gain would be expected since it is well established that zinc bacitracin improves growth performance [4]. This hypothesis is supported by studies showing that supplementation of ASA diets improved productive performance and physiological traits under heat stress conditions in poultry [32,52,53,54] and Japanese quails [53]. On the other hand, no differences in performance parameters were detected in broiler chickens treated or not with SS under non-stress conditions [17].

Tibial dyschondroplasia and footpad dermatitis are pathological conditions commonly faced in commercial poultry and are associated with many factors, and both diseases reduce overall welfare, technical performance, and carcass yield in broiler chickens [55,56]. Since studies have shown that anti-inflammatory substances ameliorate the performance of birds in these conditions [57,58], we also evaluated whether the continuous use of SS would improve the welfare of birds affected by tibial dyschondroplasia and footpad dermatitis. However, since the experiment was conducted with a small number of animals and under adequate housing conditions, no abnormalities were detected in any of the animal groups.

## 5. Conclusions

Gathering data from these toxicological evaluations, we can assume that doses of SS up to 90 mg/kg administered daily for 42 days caused a reduction in haemoglobin concentration as the only adverse effect. Nevertheless, studies are needed to verify the relevance of this finding. Moreover, sodium salicylate supplementation was not significantly efficient in promoting performance improvement; however, not even the birds treated with ZB showed improvement in performance when compared to those of the control group. In this way, it is possible that the conditions under which the birds were raised were not stressful enough to allow for the possible positive effects of both SS and ZB to be highlighted as growth promoters. Thus, studies should be conducted in which the animals are exposed to commercial conditions, where natural stressors could better evaluate the anti-inflammatory efficacy on performance.

## Figures and Tables

**Table 1 animals-13-01430-t001:** Composition and calculated nutritional values of the experimental diets of broilers treated for 42 days.

	Diets	
Parameter	Starter (Days 1–21)	Grower-Finisher (Days 22–42)
Ingredients, %		
Ground Corn	48.84	56.66
Soybean meal (45%)	40.85	33.28
Soy oil	4.61	4.67
Dicalcium phosphate	1.63	1.43
Calcitic limestone	0.86	0.73
Mineral supplement ^1^	0.10	0.10
Vitamin supplement ^2^	0.08	0.08
DL-Methionine	0.16	0.14
L-Lysine	0.16	0.21
L-Threonine	0.06	0.07
Salt	0.45	0.43
Inert filler ^3^	2.2	2.2
TOTAL	100	100
Calculated nutritional values		
Metabolisable energy, kcal/kg	3.050	3.150
Crude protein, %	23.31	20.58
Lysine ^4^ (%)	1.256	1.124
Methionine ^4^ (%)	0.515	0.461
Threonine ^4^ (%)	0.829	0.742
Valine ^4^ (%)	0.967	0.865
Phosphorus ^4^ (%)	0.419	0.374
Calcium (%)	0.878	0.758
Sodium (%)	0.221	0.211

^1^ Guaranteed minimum nutritional values per kg diet: Copper (min) 6.3 g; Iron (min) 52.5 g; Iodine (min) 1.26 g; Manganese (min) 70 g; Selenium (min) 300 mg; Zinc (min) 63 g. ^2^ Guaranteed minimum nutritional values per kg diet: Folic Acid (min) 750 mg; Pantothenic Acid (min) 10 g; Biotin (min) 80 mg; Niacin (min) 40 g; Vitamin A (min) 8000 IU; Vitamin B1 (min) 3000 mg; Vitamin B12 (min) 8000 g; Vitamin B2 (min) 6000 mg; Vitamin B (min) 3250 mg; Vitamin D3 (min) 2,500,000 IU; Vitamin E (min) 15,000 IU; Vitamin K3 (min) 2500 mg. ^3^ The space that was intended for inert (sand) was partially replaced by granulate containing SS. The inclusion of granulated varied weekly and according to the treatment in order to reach the intended doses. ^4^ Digestible values.

**Table 2 animals-13-01430-t002:** Performance of broilers treated from the 1st to the 42nd day of life with diets containing sodium salicylate at doses of 0 (negative control; BD group), 10, 30, or 90 mg/kg; or diet supplemented with 55 ppm zinc bacitracin (positive control; ZB group).

Parameter/ Week	Group					SEM	*p*-Value *^¥^*		
	0 (BD)	S10	S30	S90	ZB		Linear	Quadratic	*p*-Value †
	Body weight (g)								
1 _(d 7)_	148	150	154	158	153	5.37	0.2054	0.7086	0.7218
2 _(d 14)_	406	420	434	435	422	13.38	0.1655	0.2858	0.4530
3 _(d 21)_	876	871	914	896	877	24.71	0.4459	0.2956	0.6788
4 _(d 28)_	1497	1472	1463	1491	1527	45.27	0.8984	0.6170	0.8533
5 _(d 35)_	2147 *	2191 *	2213	2227	2365	46.21	0.2343	0.4380	0.0131
6 _(d 42)_	2769	2847	2775	2803	2955	82.61	0.9809	0.9774	0.4005
	Body weight gain (g)								
1 _(d 1–7)_	102	104	109	112	110	5.28	0.1958	0.6221	0.6413
2 _(d 8–14)_	258	258	267	277	269	10.18	0.1287	0.8840	0.5876
3 _(d 15–21)_	469	461	466	472	466	12.34	0.6516	0.7373	0.9732
4 _(d 22–28)_	597	583 *	602	583 *	646	15.44	0.6653	0.6595	0.0397
5 _(d 29–35)_	686	719	663	690	801	33.35	0.8435	0.5890	0.0529
6 _(d 36–42)_	596	656	572	582	662	83.08	0.6904	0.8628	0.8604
D1 to 21 ^1^	830	825	869	850	834	24.66	0.4364	0.2822	0.6680
D22 to 42 ^1^	1888	1958	1877	1895	2060	96.27	0.8563	0.9009	0.1753
D1 to 42	2725	2802	2730	2757	2913	82.91	0.9837	0.9742	0.3912
	Feed intake (g) ‡								
1 _(d 1–7)_	139 *	107	111	109	111	7.09	0.0227	0.0114	0.0084
2 _(d 8–14)_	327	321	321	335	333	12.62	0.3829	0.5309	0.7660
3 _(d 15–21)_	592	582	583	578	575	13.02	0.5450	0.7858	0.9094
4 _(d 22–28)_	901 *	929	1000	929	981	16.91	0.5249	0.0001	0.0005
5 _(d 29–35)_	1299	1323	1275	1283	1331	48.66	0.6614	0.6249	0.6727
6 _(d 36–42)_	1146	1095	974	912	1000	67.56	0.0147	0.2589	0.0918
D1 to 21^1^	1057	1039	1052	1022	1019	17.12	0.1499	0.7569	0.3103
D22 to 42^1^	3379	3295	3231	3160	3312	92.96	0.0930	0.4936	0.4654
D1 to 42	4440	4319	4260	4177	4333	96.37	0.0761	0.4372	0.3490
	Feed:Gain ratio								
1 _(d 1–7)_	1.376 *	1.038	1.061	0.983	1.014	0.06	<0.0001	0.0031	0.0002
2 _(d 8–14)_	1.266	1.251	1.236	1.214	1.230	0.02	0.0422	0.5884	0.2895
3 _(d 15–21)_	1.265	1.272	1.258	1.249	1.259	0.04	0.6759	0.9808	0.9953
4 _(d 22–28)_	1.514	1.600	1.645	1.599	1.527	0.04	0.3534	0.0574	0.0966
5 _(d 29–35)_	1.804	1.805	1.884	1.759	1.678	0.11	0.3853	0.2482	0.4901
6 _(d 36–42)_	2.084	1.822	1.984	1.893	1.686	0.28	0.7873	0.8873	0.8389
D1 to 21 ^1^	1.277	1.220	1.216	1.210	1.232	0.03	0.1595	0.2177	0.3693
D22 to 42 ^1^	1.811	1.704	1.734	1.691	1.617	0.08	0.4221	0.6843	0.4727
D1 to 42	1.650	1.551	1.566	1.526	1.495	0.05	0.1673	0.4519	0.2894

BD: Basal diet (without addition of Zinc Bacitracin or SS); ZB: Diet with the addition of 55 ppm Zinc Bacitracin; SEM: standard error of the mean. ^1^ Until the 21st day of the experiment *n* = 14 per group. From the 22nd to the 42nd day *n* = 9 per group. * Mean significantly different from the mean of the ZB group at this particular time (*p* ≤ 0.05). ***^¥^***
*p*-values were obtained from polynomial tests (only for Sodium salicylate), *p* ≤ 0.05 was considered significant. † *p*-values were obtained from ANOVA with Dunnett’s post-test, considering the ZB group as the reference treatment, *p* ≤ 0.05 was considered significant. ‡ There was a significant effect for the time × treatment interaction (*p* ≤ 0.05).

**Table 3 animals-13-01430-t003:** Haematological parameters of broilers treated from the 1st to the 42nd day of life with diets containing sodium salicylate at doses of 0 (negative control; BD group), 10, 30, or 90 mg/kg; or diet supplemented with 55 ppm zinc bacitracin (positive control; ZB group).

	Parameters											
Groups	RBC	HB	HT	MCV	MCH ^‡^	MCHC	WBC	Mon	Lym	Het	Eos	H:L
21 days (*n* = 12)												
BD	1.79	6.69	32.7	184.9	37.8	20.3	18,583	347	10,323	7036	157	0.68
S10	1.88	6.48	30.8	164.8	33.7	20.7	19,545	173	10,518	8155	189	0.78
S30	1.89	6.44	31.7	168.1	34.3	20.6	18,200	298	9990	7680	102	0.80
S90	1.78	5.84 *	31.5	174.2	33.3 *	18.5 *	18,500	257	11,379	6335	97	0.53
ZB	1.74	7.16	31.2	176.1	40.8	22.5	19,417	169	10,766	7901	161	0.65
SEM	0.06	0.283	0.74	6.31	2.18	1.09	1250.5	90.8	922.0	771.3	58.2	0.07
*p*-value †	0.2091	0.0096	0.3566	0.1248	0.0327	0.0253	0.9076	0.5193	0.8345	0.3846	0.7240	0.0652
Linear ^¥^	0.4174	0.0158	0.6532	0.7474	0.2349	0.0662	0.7423	0.8395	0.3452	0.1630	0.2927	0.0272
Quadratic ^¥^	0.1592	0.9484	0.4546	0.0704	0.3922	0.4676	0.4317	0.8197	0.5555	0.3124	0.6342	0.0568
42 days (*n* = 9)												
BD	2.39	9.24	33.1	138.9	38.7	28.0	23,500	1883	11,116	7706	869	0.56
S10	2.32	9.06	32.3	139.9	39.2	28.1	21,857	1657	10,077	8199	648	0.72
S30	2.39	9.19	33.3	140.2	38.6	27.4	21,444	1400	10,211	7526	487	0.69
S90	2.34	7.99 *	32.0	137.5	34.4 *	25.1 *	20,000	813	8512	7796	571	0.70
ZB	2.45	8.80	32.0	131.1	36.1	27.4	20,444	932	11,420	7649	443	0.66
SEM	0.07	0.28	1.04	5.10	1.46	0.45	2434.3	360.3	1131.9	1094.7	182.3	0.09
*p*-value †	0.7052	0.0074	0.7631	0.6096	0.0621	<0.0001	0.8315	0.1639	0.3433	0.9932	0.2583	0.7318
Linear ^¥^	0.7879	0.0006	0.4899	0.7357	0.0076	<0.0001	0.3380	0.2228	0.1001	0.9370	0.3184	0.5159
Quadratic ^¥^	0.8785	0.2603	0.6457	0.7585	0.3910	0.4065	0.7434	0.7773	0.9558	0.8743	0.1326	0.3971

RBC: Red blood cell count (×10^6^/µL); HB: Haemoglobin (g/dL); HT: haematocrit (%); MCV = mean corpuscular volume (fL); MCH = mean corpuscular haemoglobin (pg); MCHC = mean corpuscular haemoglobin concentration (g/dL); WBC: White blood cell count (/µL); Mon: monocytes (/µL); Lym: lymphocytes (/µL); Het: heterophils (/µL); Eos: eosinophils (/µL); H/L: heterophil to lymphocyte ratio; SEM: Standard error of mean. * Mean significantly different from the mean of the ZB group at this time (*p* ≤ 0.05). ^¥^
*p*-values were obtained from polynomial tests (only for Sodium salicylate), *p* ≤ 0.05 was considered significant. † *p*-values were obtained from ANOVA with Dunnett’s post-test, considering the ZB group as the reference treatment, *p* ≤ 0.05 was considered significant ‡ There was a significant effect for the time × treatment interaction (*p* ≤ 0.05).

**Table 4 animals-13-01430-t004:** Serum biochemical parameters of broilers treated from the 1st to the 42nd day of life with diets containing sodium salicylate at doses of 0 (negative control; BD group), 10, 30, or 90 mg/kg; or diet supplemented with 55 ppm zinc bacitracin (positive control; ZB group).

Groups	Parameters											
	ALT	AST	LDH	GGT	ALP	TP	ALB	GLOB	GLU	CK	UA	CR
21 days (*n* = 12)												
BD	17.6	597.4	4288 *	16.06	9118	3.133	1.81	1.325	259	4429	6.895	0.279
S10	13.8	432.4	3496	20.42	7797	3.308	1.84	1.491	271	2764	6.982	0.284
S30	16.7	478.9	2989	17.07	6418	3.625	1.93	1.470	264	3471	4.869	0.340
S90	13.8	446.4	3017	16.57	6809	3.558	1.90	1.310	278	3584	5.724	0.274
ZB	13.2	422.1	2989	16.92	6945	3.327	1.93	1.427	277	2662	6.271	0.296
SEM	1.7621	71.220	366.32	3.1233	1002.93	0.1936	0.0731	0.0841	6.9106	813.77	0.9386	0.0303
*p*-value †	0.2699	0.2786	0.0384	0.8535	0.4410	0.0870	0.6061	0.3930	0.2218	0.4397	0.1975	0.5505
Linear ^¥^	0.2698	0.2984	0.0499	0.7356	0.3631	0.1147	0.3404	0.4200	0.1237	0.8556	0.0886	0.7979
Quadratic ^¥^	0.9108	0.3675	0.0664	0.7923	0.3164	0.1097	0.2802	0.1628	0.8900	0.4165	0.1471	0.1144
42 days (*n* = 9)												
BD	10.7	377.4	2546	11.68	3349	3.125 *	1.937	1.187	243	2290.4	4.704	0.278
S10	11.6	394.2	2238	12.66	2886	3.529	2.086	1.443	260	2795.8	4.823	0.257
S30	5.4	362.9	1951	11.22	6431	3.589	2.078	1.511	258	1843.4	3.444	0.244
S90	6.5	316.6	1800	12.69	4889	3.656	2.033	1.622	270 *	1633.1	4.170	0.256
ZB	8.4	372.1	1890	10.72	5405	3.462	2.022	1.400	245	2454.2	4.003	0.233
SEM	1.7209	24.2060	301.82	1.8782	1172.94	0.0996	0.07042	0.0893	7.1525	458.21	0.5362	0.0272
*p*-value †	0.0499	0.1595	0.2889	0.9013	0.4107	0.0024	0.0857	0.0101	0.0220	0.2875	0.2991	0.7546
Linear ^¥^	0.0472	0.0111	0.0919	0.7589	0.4389	0.0019	0.1570	0.0022	0.0142	0.0802	0.4102	0.6508
Quadratic ^¥^	0.0838	0.7948	0.3236	0.7275	0.3154	0.0086	0.1030	0.0688	0.4026	0.6777	0.1007	0.3586

ALB: Albumin (U/L); ALT: Alanine aminotransferase (U/L); AST: Aspartate aminotransferase (U/L); UA: Uric acid (mg/dL); CK: Creatine kinase (U/L); CR: Creatinine (mg/dL); ALP: Alkaline phosphatase (U/L); GGT: Gamma glutamyl transferase (U/L); GLU: Glucose (U/L); GLOB: Globulin (g/dL); LDH: Lactate dehydrogenase (U/L); TP: Total protein (g/dL). ZB: Diet with the addition of 55 ppm zinc bacitracin; SEM: standard error of mean. * Mean significantly different from the mean of the ZB group at this particular time (*p* ≤ 0.05). ^¥^
*p*-values were obtained from polynomial tests (only for Sodium salicylate), *p* ≤ 0.05 was considered significant. † *p*-values were obtained from ANOVA with Dunnett’s post-test, considering the ZB group as the reference treatment, *p* ≤ 0.05 was considered significant.

## Data Availability

Not applicable.

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
