# Peer review of "Sodium Salicylate as Feed Additive in Broilers: Absence of Toxicopathological Findings"

_animals, 2023, doi:10.3390/ani13091430_

Round 1

Reviewer 1 Report

The quality of work is commendable but proofreading your paper will help eliminate grammatical  errors  which makes some portion hard to follow.

i.e abstract ln 35

is it commercial breeding  or commercial conditions ?

Methods

Line 103...can recovery of SS be above 100%

Table 1. is it 48,84 or 48.84% 

Ln. 151 21 and 42 degrees ???

Author Response

Answer to queries reviewer:

Dear editor and reviewers, we are resubmitting the manuscript with the necessary corrections requested. Thank you for the reviews and suggestions. We are very grateful for the reviewers' collaboration.

Reviewer 1:

Point 1: The quality of work is commendable but proofreading your paper will help eliminate grammatical errors which makes some portion hard to follow.

i.e abstract ln 35 is it commercial breeding or commercial conditions?

Response 1: Thank you very much for the compliment. The work was reviewed and corrected with regard to the mentioned errors. Attached, please find the certificate of English correction.

In the abstract, line 35 was corrected to “commercial conditions”. Thank you for the correction.

Methods

Point 2: Line 103...can recovery of SS be above 100%

Response 2: Dear reviewer, the recovery of 106.7% is within the acceptable range for pharmaceutical analysis and may be due to the analytical methodology and the preparation process. For example, the monograph of Aspirin tablets shown in the US Pharmacopeia 30ed allows a range of 10% of the active pharmaceutical ingredient (API) labeled in the product. This range of 10% may be found in several other monographs of the marketed products. Reference: The United States Pharmacopeia 30th Edition and National Formulary 25th Edition.  Author: United States Pharmacopeial Convention. Print Book, English, 2006.

Point 3: Table 1. is it 48,84 or 48.84%

Response 3:Thank you for the contribution. The alteration was done, it is 48.84%.

Point 4: Ln. 151 21 and 42 degrees???

Response 4: Thank you for the correction. The alteration was done for the 21st and 42nd day.

Reviewer 2 Report

The submitted manuscript “Sodium salicylate as feed additive in broilers: toxicopathological findings” focus on a very pertinent issue, since the use of antimicrobials as growth promotors is a threat to human health.

The Tittle of the manuscript could be improved to reflect the specific content of the study. Toxicopathologic findings doesn´t seem adequate since no characteristic adverse effects of NSAIDs or other toxicologic findings were observed in birds from any of the treatment groups. For example, “Inefficiency of sodium salicylate as feed additive in broilers” or “Sodium salicylate as feed additive in broilers: absence of toxicopathological findings” could be a better tittle.

The Abstract seems satisfactory and well written.

The keywords should be more adequate to the content of the study.

The introduction gives clear and enough information to understand the importance and relevance of the study.

The methodology seems appropriate, although the individual housing of the broiler chicks does not correspond to the real industrial production conditions, which could influence the results.

The results are correctly exposed except the data not shown. The discussion is adequate, although it seems that part of the results were already stated in another study of the same authors - Almeida, E.R.M.; Górniak, S.L.; Di Gregorio, M.C.; Araújo, C.S.S.; Andréo-Filho, N.; Momo, C.; Hueza, I.M. Safety and growth-481 promoting potential of repeated administration of sodium salicylate to broilers. Animal-Open Space 2022, 1, 100026. 482 https://doi.org/10.1016/j.anopes.2022.100026

The conclusions are less relevant since only stated the absence of toxicological effects of SS administrated to broilers.

Author Response

Dear editor and reviewers, we are resubmitting the manuscript with the necessary corrections requested. Thank you for the reviews and suggestions. We are very grateful for the reviewers' collaboration.

Reviewer 2:  

The submitted manuscript “Sodium salicylate as feed additive in broilers: toxicopathological findings” focus on a very pertinent issue, since the use of antimicrobials as growth promotors is a threat to human health.

Point 1: The Tittle of the manuscript could be improved to reflect the specific content of the study. Toxicopathologic findings doesn´t seem adequate since no characteristic adverse effects of NSAIDs or other toxicologic findings were observed in birds from any of the treatment groups. For example, “Inefficiency of sodium salicylate as feed additive in broilers” or “Sodium salicylate as feed additive in broilers: absence of toxicopathological findings” could be a better tittle.

Response 1: Thank you for the suggestion. The title has been changed to “Sodium salicylate as feed additive in broilers: absence of toxicopathological findings”.

The Abstract seems satisfactory and well written.

Answer: Dear reviewer, thank you very much for your comments and approvals.

Point 2: The keywords should be more adequate to the content of the study.

Response 2: Thank you for the suggestion. The alteration was done.

The introduction gives clear and enough information to understand the importance and relevance of the study.

Point 3:The methodology seems appropriate, although the individual housing of the broiler chicks does not correspond to the real industrial production conditions, which could influence the results.

Response 3: Dear reviewer, for this initial study, we opted to administer the treatment individually to ensure accurate dosing for each animal. However, further studies are already underway, where we are conducting zootechnical evaluations in commercial farming conditions to assess performance. These studies will provide more insight into the efficacy of the treatment and help determine the optimal dosage for group administration.

Point 4: The results are correctly exposed except the data not shown. The discussion is adequate, although it seems that part of the results were already stated in another study of the same authors - Almeida, E.R.M.; Górniak, S.L.; Di Gregorio, M.C.; Araújo, C.S.S.; Andréo-Filho, N.; Momo, C.; Hueza, I.M. Safety and growth-481 promoting potential of repeated administration of sodium salicylate to broilers. Animal-Open Space 2022, 1, 100026. 482 https://doi.org/10.1016/j.anopes.2022.100026.

Response 4: Dear reviewer, we chose not to show the histopathological data and the tibial changes and footpad dermatitis due to the absence of significant changes when compared with the control group animals.

Although the present study builds upon our group's previous research (Almeida et al., 2022), it employs a different methodology that includes higher doses of sodium salicylate (SS) and additional assessments, such as footpad dermatitis evaluation. As a result, the experimental design has undergone significant changes, and the inclusion of new assessment allows for a more comprehensive investigation of the potential toxicological effects of SS.

Point 5: The conclusions are less relevant since only stated the absence of toxicological effects of SS administrated to broilers.

Response 5:  In conclusion, our findings reinforce that the dose of sodium salicylate used in this study did not produce any toxic effects in the animals. These results suggest that the use of SS in animal production may be a safe option for managing certain conditions, and further studies are warranted to investigate the improvement in zootechnical indices.

Reviewer 3 Report

The manuscript is interesting, up-to-date, and very well written. The Introduction provides sufficient background and justification of the research. Material and methods does not raise any doubts. The properly performed hematological analysis of the birds' blood and the statistical analysis of the obtained results deserve to be emphasized. The results are clearly described and discussed. The manuscript definitely deserves publication in Animals.

There are only a few things that have to be corrected before publication:

Table 1, row “Inert” – Please, specify what was used as the “Inert” in basal diet. Was it cornmeal?

Table 1’s legend – The explanation of no. 3 is a little bit confusing. Please define the “aggregate” and provide more precise explanation for the “Inert” in diets with sodium salicylate.

L. 151 – There should be 21st and 42nd day.

L. 217 – Throughout the whole manuscript, please unify the abbreviation used for a diet with zinc bacitracin (ZB or BZ).

L. 222 – At week 4, ZB increased feed intake as compared to the BD group. Please, correct this.

Table 2, 3, and 4 – Please use the same order of groups in each table.

Table 2’s legend – The superscript numbers 1 and 2 do not refer to anything in Table 2.

L. 245-247 and Table 3 – There should be MCH for mean corpuscular hemoglobin.

L. 255 – There should be GLU for glucose.

Table 3 – There is an asterisk (*) missing in Table 3 for MCH in the S90 group after 42 days of the trial (34.4 was significantly different from 36.1 in ZB group).

Table 3’ legend – A unit should be added for Lym, Mon, Het, and Eos. It can be assumed that the results of leukogram analysis are expressed in percentage but it has to be specified. The H/L should be defined as heterophils to lymphocytes ratio.

L. 281 – There should be “Lactate dehydrogenase”.

L. 306-309 – The description is confusing. It can be understand that most birds presented lesions. Please rewrite this paragraph.

L. 311 – There should be “The gastrointestinal tract”.

L. 406-407 – The last sentence can be deleted.

L. 410 – The SS was administered for 42 days, and no longer, so please correct this.

Author Response

Dear editor and reviewers, we are resubmitting the manuscript with the necessary corrections requested. Thank you for the reviews and suggestions. We are very grateful for the reviewers' collaboration.

Reviewer 3:

Point 1: The manuscript is interesting, up-to-date, and very well written. The Introduction provides sufficient background and justification of the research. Material and methods does not raise any doubts. The properly performed hematological analysis of the birds' blood and the statistical analysis of the obtained results deserve to be emphasized. The results are clearly described and discussed. The manuscript definitely deserves publication in Animals.

Response 1: Dear reviewer, we are very grateful for your gentleness and compliments.

There are only a few things that have to be corrected before publication:

Point 2: Table 1, row “Inert” – Please, specify what was used as the “Inert” in basal diet. Was it cornmeal?

Response 2: The inert used was sand.

Point 3: Table 1’s legend – The explanation of no. 3 is a little bit confusing. Please define the “aggregate” and provide more precise explanation for the “Inert” in diets with sodium salicylate.

Response 3: Thank you for the correction. The correct term is inert. The space that was intended for inert (sand) was partially replaced by granulate containing SS. The inclusion of granulated varied weekly and according to the treatment in order to reach the intended doses.

Point 4: L. 151 – There should be 21st and 42nd day.

Response 4: Thank you for the correction. The alteration was done for the 21st and 42nd day.

Point 5: L. 217 – Throughout the whole manuscript, please unify the abbreviation used for a diet with zinc bacitracin (ZB or BZ).

Response 5: Thank you for the correction. It was already reviewed and changed in the text to ZB.

Point 6: L. 222 – At week 4, ZB increased feed intake as compared to the BD group. Please, correct this.

Response 6: Thank you for the correction. It was already changed.

Point 7: Table 2, 3, and 4 – Please use the same order of groups in each table.

Response 7: Thank you. It was corrected.

Point 8: Table 2’s legend – The superscript numbers 1 and 2 do not refer to anything in Table 2.

Response 8: Thank you for the correction. It was done.

Point 9: L. 245-247 and Table 3 – There should be MCH for mean corpuscular hemoglobin.

Response 9: Thank you for the correction. It was done.

Point 10: L. 255 – There should be GLU for glucose.

Response 10: Thank you for the correction. It was done.

Point 11: Table 3 – There is an asterisk (*) missing in Table 3 for MCH in the S90 group after 42 days of the trial (34.4 was significantly different from 36.1 in ZB group).

Response 11: Thank you. It was corrected.

Point 12: Table 3’ legend – A unit should be added for Lym, Mon, Het, and Eos. It can be assumed that the results of leukogram analysis are expressed in percentage but it has to be specified. The H/L should be defined as heterophils to lymphocytes ratio.

Response 12: Thank you for the correction. The units were inserted and the text was corrected.

Point 13: L. 281 – There should be “Lactate dehydrogenase”.

Response 13: Thank you. It was corrected. 

Point 14: L. 306-309 – The description is confusing. It can be understand that most birds presented lesions. Please rewrite this paragraph.

Response 14: Thank you. The paragraph was rewritten.

Point 15: L. 311 – There should be “The gastrointestinal tract”.

Response 15: Thank you for the correction.

Point 16: L. 406-407 – The last sentence can be deleted.

Response 16: Thank you for the suggestion. It was done.

Point 17: L. 410 – The SS was administered for 42 days, and no longer, so please correct this.

Response 17: Thank you. It was corrected.

Round 2

Reviewer 2 Report

The major concerns were corrected by the authors.